# Epidemiology of unintentional childhood injuries in urban and rural areas of Nepal- A comparative study

**Pratiksha Pathak** [1]*, **Sunil Kumar Joshi** [2]

1 Department of Community Health Sciences, Patan Academy of Health Sciences, Lalitpur, Nepal,
2 Department of Community Medicine, Kathmandu Medical College, Kathmandu, Nepal

* pratikshapathak@pahs.edu.np, pratiksha.pthk3@gmail.com

**Data Availability Statement:** All relevant data are within the manuscript and its Supporting information files.

**Funding:** This research was funded by Nepal Health and Research Council as "Post Graduate

## Abstract

Unintentional injuries are one of the leading causes of ill health, disability & death among the children and young adults worldwide. As these injuries are strongly related to social determinants, the burden falls mainly upon the Low- and Middle-Income Countries (LMICs) like Nepal. Thus, the main objective was to explore the epidemiology of unintentional childhood injuries in urban and rural areas of Kavrepalanchok district. A cross sectional analytical study was done in Bethanchok rural municipality and Dhulikhel municipality of Kavrepalanchok district. The respondents were interviewed using a pretested semi-structured questionnaire. The details of injuries sustained within the past 12 months were included. A total of 667 children aged 1–16 years were surveyed, among which 26% from rural and 17.2% from urban areas had unintentional injuries in the past 12 months. Falls were the most common mode of injury in both the areas. Similarly, the proportion of burn was more in rural area (16.1%) whereas, Road Traffic Injuries were more in urban area (12.5%). Majority of the injuries occurred at home (54.5%) while the child was playing (64.1%). Factors like child's gender and place of residence affected the occurrence of unintentional injuries (p<0.05). Out of total injured children, 18 of them had not recovered and 11 were left with some form of permanent disability. As the rate and pattern of unintentional childhood injuries in the rural and urban area differ, the prevention strategies should focus on risk factors that apply to both the areas and awareness should be created among the parents and primary caregivers about the fact that childhood injuries are predictable and preventable.

## Introduction

Around the world, childhood injury is a major public health problem affecting both developed and developing countries. As per World Health Organization (WHO), Road Traffic Injuries (RTIs) stand at the top three causes of deaths among 5–29-year-olds and drowning being the sixth leading cause of deaths among 5–14-year-olds. Similarly, falls are accountable for more than 68400 deaths annually [1]. Since the burden of infectious diseases have been reduced substantially because of prevention and early intervention techniques, unintentional injuries are emerging as a public health problem [1–5].

Health Research Grant, 2018" (ref. No. 2384). URL: https://nhrc.gov.np/ The funders had no role in study design, data collection and analysis, decision to publish, or preparation of the manuscript.

**Competing interests:** The authors have declared that no competing interests exist.

Nepal is one of the least developed countries in the world. Child mortality is still very high in Nepal, despite the reduction in rates by two-third in the past 20 years [6]. Also, Nepal is ranked among the 49 least developed countries in the world with 39% of the population falling below the poverty line and comprised of a uniquely challenging physical environment that adds on to injury risk for children [7, 8]. Nepal's Health Management Information System is the only routinely collected source of information on non-fatal injuries; this recorded 1.1 million outpatient department visits for injuries in the year 2017–2018. Also, delay in seeking medical care and thinking that 'injury will heal with time' are the common misconceptions among people [9]. There are multiple factors to be considered in understanding unintentional childhood injuries such as child's age, gender, prior injury, maternal health, maternal age, and environmental factors [10]. Maternal depression, hazardous adult alcohol use, and socioeconomic deprivation were identified as important modifiable risk factors for injuries in preschool children [11]. Injurious events were thought to be commonly preceded by hunger and tiredness of both children and parents as well as a high level of sibling interaction [10].

The most reported risk factors from SEAR were age, gender, and place of residence; Male gender and increasing age were associated with nonfatal injuries also the mental health status of caregivers was associated with all forms of injuries among infants and toddlers. While analyzing the risk factors for childhood drowning, it was revealed that increasing maternal age (above 30), maternal illiteracy and low family income were significantly associated with drowning. Similarly, aggressive behaviors of the children, police encounter and warnings for poor riding were the risks behind injuries among under-age motorcycle riders [12–15].

Also, most of the children who are victims of non-fatal injuries suffer some form of disability, notably due to injuries to extremities, brain injuries, and spinal cord injury [13, 14]. The rehabilitation of these disabled persons can be costly thus, difficult to afford as a result of poor economic resources, which highlights the importance of prevention efforts for reducing the occurrence of injuries and reducing its burden to healthcare, social and judicial services [13]. Thus, the study aims to explore epidemiology of unintentional childhood injuries in urban and rural areas of Kavrepalanchok district of Nepal for better understanding of various attributes in addressing unintentional childhood injuries.

## Methods

A cross-sectional analytical community-based study was conducted in Bethanchok rural municipality and Dhulikhel municipality of Kavrepalanchok district of Nepal. The ethical clearance was sought from Institutional Review Committee (IRC) of Kathmandu Medical College. All the participants were informed in detail about the study and written assent was taken. Children aged 1–16 years from selected households, whose parents gave consent for the study were included; children who had congenital defects or major illness were not included in the study. The respondents were their parents or primary care givers. Data collection was done by face-to-face interview between October and November 2018 using pretested semi-structured questionnaire adapted from 'Guidelines for Conducting Community Surveys on Injuries and Violence by World Health Organization'. The unintentional injuries are those where the intent to use the force may not necessarily intend to cause the damage. This issue can be quite complex as the intent to use force may not necessarily mean that there was an intent to cause harm or damage. Unintentional injuries are classified as per their occurrence: poisoning, burns, and scalds, drowning, falls and transport-related [13].

The respondents were interviewed separately to maintain confidentiality and avoid influence from outside. Also, options or hints to any question were not given and only those options they gave as answer were ticked.

Sample size was calculated as follows:
Sample size calculation was done as follows;

$$\text{Sample size } (n) = \frac{z^2 . p . q}{d^2}$$

Where,
z = at 95% confidence level (standard value = 1.96)
p = prevalence of unintentional injury in Asian children = 5.5% [16] = 0.055
q = 1-p = 1–0.055 = 0.0945
d = margin of error of 5%

$$\text{Sample Size (n)} = \frac{1.96 \times 1.96 \times 0.055 \times 0.0945}{0.05 \times 0.05} = 80$$

Considering 10% non—response rate,

$$\text{Sample size (n)} = 80/0.9 = 89$$

Considering the design effect, Sample size (n) = 89 X 4 = 356
Since, the study was conducted both in rural and urban area,

$$\text{Final sample size} = 356 \times 2 = 712$$

Thus, the final sample size was 712 considering the prevalence of unintentional injury among Asian children as 5.5% [16].

Multi-stage sampling technique was used for sample collection. Bethanchok rural municipality and Dhulikhel municipality were randomly chosen from Kavrepalanchok district. From Bethanchok and Dhulikhel, four wards were randomly chosen; further the sample to be taken from each ward was calculated using Probability Proportional to Size (PPS) method. Finally, the households required to reach the target sample size in each ward were selected using convenience sampling method. Only 1 child was taken from single household; in case of more than 1 child, the youngest child above 1 years was only included Fig 1.

The collected data was entered in MS Excel and statistical analysis was done using Statistical Package for Social Sciences (SPSS). The descriptive statistical tools like frequency, percentage, mean, standard deviation, median and graphs were used to express the results. Pearson chi square test was used for bivariate analysis to determine the association between independent and dependent variables. Crude odds ratio (COR) at 95% confidence interval (CI) was calculated to see the magnitude of association with independent variables. In multivariate analysis, binary logistic regression was carried out. Independent variables associated with unintentional childhood injuries with a significance level of less than 0.2 (p<0.2) were adjusted among themselves in multivariate logistics regression to calculate the Adjusted Odds Ratio (AOR). Those independent variables that reached statistically significant level (p-value <0.05) were considered as important factors affecting the unintentional injuries.

## Results

### Sociodemographic features

The overall response rate was 93.7% (rural: 96.0% and urban: 91.3%). Thus, the total number of participants in this study were 667 out of which 342 were from rural area and 325 from urban area. About two-third of the respondents were mothers (64.9%) followed by fathers (32.7%) and few of them were grandparents (2.4%). The age of the children ranged from 1–16

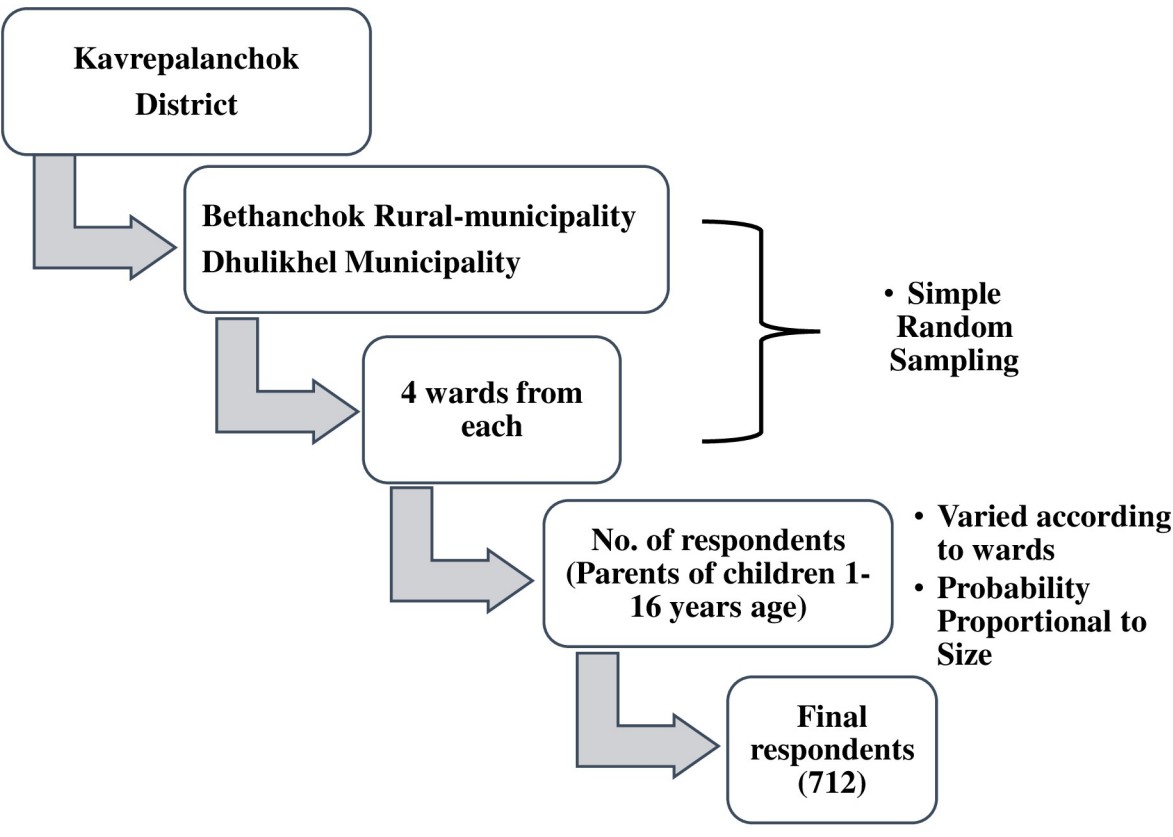

**Fig 1. Flow diagram of the sampling technique.**

years in both the areas. The median age of children in rural area was 5 years with an inter-quartile range of 6 years and in urban area was 7 years with an inter-quartile range of 7 years.

Similarly, the details of 658 mothers (335 and 323 in rural and urban area respectively) and 659 fathers (337 and 322 in rural and urban area respectively) were included. Total 9 mothers and 8 fathers were either separated or expired thus they were excluded from the study. The socio-economic status was calculated using modified Kuppuswamy's Socioeconomic status scale for Nepal [17]. The sociodemographic details of the study participants are shown in Table 1.

### Injury event

Out of the total 667 children, 21.7% [95% CI: 18.6–24.9] had sustained unintentional injuries in the past 12 months. In rural area, the prevalence was 26% [95% CI: 21.3–30.7] whereas in urban area it was 17.2% [95% CI: 13.1–21.4].

Falls were most common mechanism of injury in both rural (65.9%) and urban (62.5%) areas followed by cut injuries. The proportion of burn cases was nearly double in rural as that of urban area whereas RTIs were more in urban area. Similarly, animal related injuries were almost equal in both the areas Table 2.

More than half of the injuries occurred at home and its premises in both the areas (rural: 58.4% and urban: 48.2%) whereas, just a proportion of injuries in had occurred near a water source (2.1%). Similarly, in majority of the cases, the child was playing while the injury had

Table 1. Sociodemographic details of participants (n = 667).

| Variable | Rural N (%) n = 342 | Urban N (%) n = 325 |
|---|---|---|
| **Age (years)** | | |
| 1–5 | 198 (57.9) | 133 (40.9) |
| 6–10 | 79 (23.1) | 116 (35.7) |
| 11–16 | 65 (19.0) | 76 (23.4) |
| **Gender** | | |
| Male | 172 (50.3) | 171 (52.6) |
| Female | 170 (49.7) | 154 (47.4) |
| **Family Type** | | |
| Nuclear | 190 (55.6) | 162 (49.8) |
| Joint | 152 (44.4) | 163 (50.2) |
| **Religion** | | |
| Hindu | 200 (58.5) | 299 (92.0) |
| Buddhist | 141 (41.2) | 24 (7.4) |
| Muslim | 1 (0.3) | 0 |
| Christian | 0 | 2 (0.6) |
| **Ethnicity** | | |
| Dalit | 11 (3.2) | 12 (3.6) |
| Janajati | 147 (43.0) | 151 (46.5) |
| Brahmin | 175 (51.2) | 137 (42.2) |
| Chhetri | 9 (2.6) | 25 (7.7) |
| **Socioeconomic status** | | |
| Upper class | 0 | 35 (10.8) |
| Upper middle class | 9 (2.6) | 207 (63.7) |
| Lower middle class | 196(57.3) | 80 (24.6) |
| Upper lower class | 119 (34.8) | 3 (0.9) |
| Lower class | 18 (5.3) | 0 |

occurred (64.1%) and, least proportion of injuries had occurred while the child was with his/her parent (15.1%). The further details on injury event are shown in Table 3.

Majority of the children (84.1%, 122) received some form of treatment after the injury in both the areas. Among those who sought treatment, more than half (55.8%, 24) of them in urban area went to hospitals followed by medical shops (20.9%, 9). Similarly, in rural area,

Table 2. Distribution of various modes of injury (n = 145).

| Modes of injury* | Rural (n = 89) N (%) | Urban (n = 56) N (%) |
|---|---|---|
| Falls | 58 (65.9) | 35 (62.5) |
| Cut injuries | 16 (18.0) | 9 (16.1) |
| Burns | 14 (16.1) | 5 (8.9) |
| Road Traffic Injuries | 2 (2.2) | 7 (12.5) |
| Animal related injuries | 5 (5.7) | 5 (8.9) |
| Accidental poisoning | 3 (3.4) | 1 (1.8) |
| Choking/ Suffocation | 0 | 2 (3.6) |
| Electric shock | 0 | 2 (3.6) |

*Multiple response

**Table 3. Details of injury event (n = 145).**

| Variables | Rural (n = 89) N (%) | Urban (n = 56) N (%) |
|---|---|---|
| **Place of injury occurrence** | | |
| Home and its premises | 52 (58.4) | 27 (48.2) |
| School | 7 (7.9) | 12 (21.4) |
| Road | 11 (12.4) | 14 (25.0) |
| Field | 18 (20.2) | 1 (1.8) |
| Water source | 1 (1.1) | 2 (3.6) |
| **Child's activity at the time of injury occurrence** | | |
| Playing | 50 (56.2) | 43 (76.8) |
| Walking | 23 (25.8) | 10 (17.8) |
| Helping in household chores | 8 (9.0) | 1 (1.8) |
| Sleeping | 8 (9.0) | 2 (3.6) |
| **Person accompanying the child at the time of injury** | | |
| Parents | 17 (19.1) | 5 (8.9) |
| Family member | 18 (20.2) | 24 (42.9) |
| Friends | 21 (23.6) | 18 (32.1) |
| Alone | 33 (37.1) | 9 (16.1) |

about one third (36.7%, 29) of them sought treatment from hospitals followed by medical shops (24.1%, 19) and home (24.1%, 19).

**Relationship between various child and parental attributes with unintentional childhood injuries.** Table 4 shows the association between various child and parental factors and occurrence of unintentional injuries. The attributes that showed significant relationship with unintentional injuries has further been adjusted with each other for multivariate analysis.

Majority of the children (84.1%, 122) received some form of treatment after the injury in both the areas. Among those who sought treatment, more than half (55.8%) of them in urban area went to hospitals followed by medical shops (20.9%). Similarly, in rural area, about one third (36.7%) of them sought treatment from hospitals followed by medical shops (24.1%) and home (24.1%).

The male child had about twice the higher chances of sustaining unintentional injuries than that of female child [COR: 2.45; 95% CI: 1.66–3.62; (p<0.001)]. Significant difference was seen in between unintentional injuries and number of siblings of the child (p = 0.01). The children living in urban area had less chances of sustaining injuries as compared to those living in rural area [COR: 0.59; 95% CI: 0.41–0.86; (p = 0.006)].

There was significant difference in unintentional injuries among the children according to the education and occupational status of their parents, their socioeconomic status and total family income (p < 0.05). The children belonging to families whose total monthly family income was less than Rs. 20000 had twice higher chances of sustaining unintentional injuries as compared to those with total monthly family income of more than Rs.50000 [COR: 2.61; 95% CI: 1.33–5.11; (p = 0.002)] Table 4.

For the multivariate analysis, various attributes of child with a significance level of less than 0.2 (p-value< 0.2) were included to calculate the adjusted odds ratio after adjusting with each other Table 4. The variables included are age, gender, number of siblings, place of residence, religion, ethnicity, educational level, occupational status of parents, socio-economic status and total monthly family income.

After adjustment, it was seen that male child had about two times higher odds of sustaining unintentional injuries as compared to female child [AOR: 2.48; 95% CI:1.67–3.69; (p<0.001)];

**Table 4. Relationship between various child and parental attributes with unintentional childhood injuries (n = 667).**

| Variables | Unintentional injuries | | |
|---|---|---|---|
| | COR [95% CI] (p-value) | | AOR [95% CI] (p-value) |
| **Age of the child (years)** | | | |
| 1–5 | 0.63 [0.39–0.99] | p = 0.12 | 0.62 [0.38–1.04] (0.07) |
| 6–10 | 0.81 [0.49–1.34] | | 0.85 [0.50–1.44] (0.54) |
| 11–16 | RC | | RC |
| **Gender of the child** | | | |
| Male | 2.45 [1.66–3.62] | p <0.001* | 2.60 [1.74–3.89] (<0.001*) |
| Female | RC | | RC |
| **Number of siblings** | | | |
| 0 | 0.49 [0.29–0.81] | p = 0.01* | 0.64 [0.36–1.13] (0.12) |
| 1 | 0.56 [0.35–0.91] | | 0.76 [0.45–1.29] (0.30) |
| >1 | RC | | RC |
| **Place of residence** | | | |
| Urban | 0.59 [0.41–0.86] | p = 0.006* | 0.57 [0.37–0.87] (0.01*) |
| Rural | RC | | RC |
| **Religion** | | | |
| Hindu | 0.93 [0.61–1.42] | p = 0.01* | 0.70 [0.33–1.47] (0.34) |
| Others | RC | | RC |
| **Ethnicity** | | | |
| Dalit | 2.97 [0.92–9.57] | p = 0.07 | 2.04 [0.53–7.82] (0.30) |
| Janajati | 0.95 [0.40–2.30] | | 0.55 [0.20–1.52] (0.25) |
| Brahmin | 1.10 [0.46–2.62] | | 0.85 [0.34–2.16] (0.73) |
| Chhetri | RC | | RC |
| **Education of the mother** | | | |
| Illiterate | 2.29 [1.26–4.17] | p = 0.002* | 0.68 [0.30–1.53] (0.35) |
| Primary | 2.47 [1.39–4.38] | | 0.89 [0.44–1.84] (0.77) |
| Secondary | 1.73 [1.11–2.71] | | 1.34 [0.61–2.97] (0.47) |
| Higher secondary and above | RC | | RC |
| **Education of the father** | | | |
| Illiterate | 3.58 [1.56–8.21] | p = 0.001* | 1.08 [0.68–1.72] (0.74) |
| Primary | 2.23 [1.28–3.91] | | 0.67 [0.33–1.36] (0.27) |
| Secondary | 1.40 [0.89–2.18] | | – |
| Higher secondary and above | RC | | RC |
| **Occupation of the mother** | | | |
| Profession | 0.93 [0.29–2.96] | p = 0.03* | 0.70 [0.12–4.14] (0.69) |
| Semi- profession | 0.51 [0.26–0.98] | | 0.76 [0.35–1.62] (0.47) |
| Clerical/ shop owner/ farmer | 1.33 [0.84–2.10] | | 0.85 [0.47–1.55] (0.66) |
| Skilled worker | 1.03 [0.36–2.97] | | 0.80 [0.26–2.54] (0.71) |
| Semi- skilled worker | – | | – |
| Unskilled worker | 1.56 [0.63–3.87] | | 1.24 [0.42–3.72] (0.69) |
| Unemployed | RC | | RC |
| **Occupation of the father** | | | |

(*Continued*)

**Table 4.** (Continued)

| Variables | Unintentional injuries | | |
|---|---|---|---|
| | COR [95% CI] (p-value) | | AOR [95% CI] (p-value) |
| Profession | 0.30 [0.08–1.15] | p = 0.007* | 0.46 [0.12–2.01] (0.30) |
| Semi- profession | 0.18 [0.56–0.57] | | 0.22 [0.06–0.91] (0.04) |
| Clerical/ shop owner/ farmer | 0.42 [0.14–1.30] | | 0.35 [0.10–1.27] (0.11) |
| Skilled worker | 0.28 [0.07–1.08] | | 0.23 [0.05–1.03] (0.05) |
| Semi- skilled worker | 0.78 [0.10–6.32] | | 0.92 [0.08–10.49] (0.94) |
| Unskilled worker | 0.31 [0.09–1.05] | | 0.28 [0.07–1.06] (0.64) |
| Unemployed | RC | | RC |
| **Socio-economic status** | | | |
| Lower class | 2.21[0.48–10.15] | p = 0.05 | 1.18 [0.21–6.60] (0.85) |
| Upper lower class | 1.80 [0.58–5.61] | | 0.94 [0.23–3.75] (0.92) |
| Lower middle class | 2.89 [0.99–8.47] | | 1.75 [0.48–6.37] (0.39) |
| Upper middle class | 1.71 [0.57–5.12] | | 1.62 [0.46–5.73] (0.46) |
| Upper class | RC | | RC |
| **Total monthly family income (Nepalese Rupees)** | | | |
| < 20000 | 2.61 [1.33–5.11] | p = 0.002* | 1.18 [0.21–6.60] (0.85) |
| 20000–29999 | 1.14 [0.41–3.14] | | 0.94 [0.23–3.75] (0.92) |
| 30000–39999 | 2.29 [1.39–3.78] | | 1.75 [0.48–6.37] (0.39) |
| 40000–49999 | 2.01 [1.25–3.22] | | 1.62 [0.46–5.73] (0.46) |
| ≥ 50000 | RC | | RC |

AOR: Adjusted Odds ratio, COR: Crude Odds Ratio, RC: Reference Category, CI: Confidence Interval,

* Statistically significant

similarly, the child living in urban area had about 40% less chances of sustaining unintentional injuries as compared to those living in rural area [AOR: 0.57; 95% CI: 0.38–0.87; (p = 0.01)] Table 4.

**Impact of injuries.** Among the children who were injured, 22 hospitalized for treatment, 11 were left with some form of disability and 10 of them were not able to return back to school. While inquiring about the impact upon the families, 14 households claimed a decrease in total household income, in 28 cases either parent had lost their workdays and one of the fathers from the rural area had lost his job permanently as he had to take care of the injured child Table 5.

## Discussion

The main aim of the study was to explore the epidemiology of unintentional injuries in urban and rural areas of Kavrepalanchok district. It was seen that out of the total children in this study, 26% from rural and 17.2% from the urban area had sustained some form of unintentional injury in the past 12 months. The finding was consistent with the findings from Makwanpur district of Nepal, South India and South Africa where the prevalence of unintentional injury was higher in rural than the urban area [7, 18–20]. Such differences might be due to differences in environmental, infrastructural, economic and cultural related factors in both the areas [19, 20].

However, the prevalence of unintentional injury was lower in Tamil Nadu (12.9%) than that of this study whereas it was higher in a similar study from Dharan, Nepal [18, 21]. The

**Table 5. Impact of injuries upon the child and their families (n = 145).**

| Impacts of injury* | Rural (n = 89) | Urban (n = 56) | Total (n = 145) |
|---|---|---|---|
| **Upon the child** | | | |
| Not Recovered | 15 | 3 | 18 |
| Hospitalization | 16 | 6 | 22 |
| Operative intervention | 8 | 4 | 12 |
| Impairment | 11 | 1 | 12 |
| Physical disability | 10 | 1 | 11 |
| Not able to return to school | 8 | 2 | 10 |
| **Upon the family** | | | |
| Lost days of work | 20 | 8 | 28 |
| Lost job | 1 | 0 | 1 |
| Decline in household income | 12 | 2 | 14 |
| Borrow money | 5 | 0 | 5 |

*Multiple response

studies from Bangladesh and Thailand also reported injury rates lower than this study which might be due to the difference in case definitions. As the treatment options are limited in a country like Nepal, the injury cases included in this study were regardless of their treatment status whereas, in Bangladesh and Thailand those injuries were only included where treatment was sought [7, 21].

The study reveals that falls were the most common mechanism of injury in both rural (65.9%) and urban (62.5%) areas. As per CDC, falls were the leading cause of non-fatal injury among all age groups of children in the US [20]. Similarly, falls comprised of more than half of total injuries in Pakistan, Tamil Nadu, urban and rural areas of Ujjain, India and Makwanpur, Nepal [7, 19, 22]. More than half of the children in this study had fallen from stairs which could have been prevented by simply ensuring railings for staircases and parapet walls in terraces [9]. However, studies from Bangladesh and China revealed that drowning was the main cause of childhood mortality which might be due to the fact that households in these areas are surrounded by numerous ponds, ditches and rivers [23, 24]. A large number of children in the household and lack of parental supervision while playing also poses a risk of drowning in the children of these regions.

In this study, RTIs were more common in urban area (12.5%) which may be due to low traffic volume in rural area whereas, burns were more among the rural children (16.1%) which was probably because rural children especially girls were more involved in household chores like cooking, cattle feed preparation. This concurs with the findings from studies conducted in Australia, Canada, Ireland, USA and South Africa where traffic related injuries and burns were more common in rural areas. Also, poor housing conditions, lack of clearly demarcated areas for cooking and use of open fires poses as a risk for burns among children [20].

Accidental poisoning was more common in the rural area of this study mainly due to ingestion of pesticides that was like the findings from a study conducted among rural and urban children of Ujjain, India [19]. This may be because the use of pesticides is more in rural areas and the storage is not proper. Thus, children can access these pesticides easily leading to accidental poisoning. The proper storage of poisoning substances in closed vessels inside locked cupboards and out of the reach of children would be effective in the prevention of poisoning to some extent.

Among all the injuries in both the areas, nearly half of them occurred at home and its premises in this study. Similarly, childhood injury surveillance from Bangladesh, Colombia, Egypt, and Pakistan revealed that 56% of the injury occurred at home [25]. Likewise, a house-hold survey in Nepal, also revealed similar findings; 39% of the childhood injuries took place in the home environment [7]. This may be because children spend most of their time in and around home and its premises which reflects the need of home-based injury prevention strategies.

Male child had nearly two times higher chances of sustaining unintentional injuries as compared to the female child (p<0.001) in this study. Various studies from Karachi, Ujjain, and other SEA countries also revealed higher injury rates among male children than females [12, 19, 22]. Similarly, in Makwanpur, Nepal also the injury rate among boys (32.5/1000) was almost double than that of girls (16.8/1000) [7]. This may be due to behavioral differences among male and female children. The restless nature of boys makes it difficult to supervise and control them than the girls of same age group [23]. Also the patriarchal nature of societies in LMCIs allow the male child to explore their environment at an early age increasing the risk of injuries [26].

This study showed that children residing in the rural area have higher chances of sustaining unintentional injuries as compared to those residing in the urban area (p = 0.01). The findings were consistent with various studies from New Zealand, urban and rural parts of India [9, 15, 18]. Also, the annual incidence of unintentional childhood injuries was higher in rural area than that of urban areas in Makwanpur, Nepal [7].

This may be due to different environmental and infrastructural factors like housing condition, overcrowding, open fields, school environment and road conditions in rural and urban areas as well as difference in parental perception on childcare and injury prevention. Thus, childhood injury prevention programs should focus on risk factors that apply to both rural and urban areas [12].

The present study shows that the proportion of impairment and disability resulting from unintentional injuries was higher in rural than in urban areas. Unintentional injuries resulted in temporary impairment of 11 children and permanent disability of 10 children from the rural area whereas very few of the urban children suffered from impairment or disability which might be because of better health care facilities in urban areas. Similarly, studies conducted in various parts of South India and Karachi showed disability proportion among the injured children to be similar to the present study [27–29].

However, the nation-wide survey among under 5 children in Pakistan showed that the proportion of disability was less than this study (3%) and the disability was highest among lowest tertile of community development probably due to lack of proper health care for the injuries [29]. Physical and psychological effects upon the children were also identified as post injury sequel by the study conducted in Makwanpur district of Nepal [30].

In this study, in nearly one-fifth of the injury cases from both the areas, either parent had lost their work days to take care of the injured child and one of the fathers from the rural area had lost his job permanently as a result of injury to the child. Likewise, the absenteeism of primary caregivers from their work from various parts of South India, to take care of injured child ranged from 1–60 days with a mean of 3 days [27, 28].

The impacts of unintentional injuries were higher in rural areas probable because of delay in seeking health care for the injury in the rural area and provision of better health care facilities urban area. However, there is a lack of national figures to estimate the burden of unintentional childhood injuries. This calls for further research both community and facility-based representing the diverse population in the country [31].

This study being a community-based study has provided the snapshot of injury burden at the community level to some extent. Due to the lack of national data, many aspects of the

study could not be compared which calls for a survey at the national level. Also, most of the data regarding injury available in our country are facility-based that masks the actual injury burden because not all the injuries seek health care at the facility. This calls for the further community-based studies on a larger scale such as household surveys, verbal autopsies, etc. to estimate the burden and establish evidence base necessary for the effective injury prevention program.

## Conclusion

This study identified the influence of various child and parental factors on the unintentional childhood injuries. It was seen that the prevalence of unintentional childhood injuries was higher in rural than in urban area. Falls were the most common mechanism of injury in both the areas and majority of the injuries took place at home and its premises. Factors like child's gender and place of residence affected the occurrence of unintentional injuries in children. This shows that the understanding of such factors is crucial in developing and implementing childhood prevention strategies.

## Strengths

This is a community-based study that might somewhat show the true burden of injuries as all the cases might not seek treatment in health facilities. Also, a comparison between rural and urban area has been done as the epidemiology of injuries is different in both places.

## Limitations

There are some limitations of this study. Even though probability sampling was used wherever possible, due to the lack of proper sampling frame, final respondents had to be selected conveniently from each ward. There might have been "Recall bias", as the respondent might not have been able to remember all the details of injury from the past 12 months. Also, proxy respondents (parents/ primary caregivers) were approached, who may not know all the information regarding the injury. And finally, the study was conducted only in one district of Nepal thus, the results cannot be generalized in case of the whole country.

## Recommendations

Further studies are required on larger scale in order to establish the evidence base necessary for effective injury prevention program. Also, establishment of both facility-based and community-based injury registry would provide further contribution for evidence generation. Similarly, childhood injury prevention programs could be introduced at school by trained teachers and also at community level by FCHVs. Furthermore, the health systems should be strengthened to address child injuries which includes the provision of high-quality care to the injured children as well as rehabilitation and support services.

## Supporting information

**S1 Data.**
(XLS)

## Acknowledgments

We would like to thank all the participants for supporting our research.

## Author Contributions

**Conceptualization:** Pratiksha Pathak, Sunil Kumar Joshi.

**Data curation:** Pratiksha Pathak.

**Formal analysis:** Pratiksha Pathak.

**Funding acquisition:** Pratiksha Pathak.

**Investigation:** Pratiksha Pathak.

**Methodology:** Pratiksha Pathak.

**Project administration:** Pratiksha Pathak, Sunil Kumar Joshi.

**Resources:** Pratiksha Pathak.

**Software:** Pratiksha Pathak.

**Supervision:** Sunil Kumar Joshi.

**Validation:** Pratiksha Pathak, Sunil Kumar Joshi.

**Visualization:** Pratiksha Pathak.

**Writing – original draft:** Pratiksha Pathak.

**Writing – review & editing:** Pratiksha Pathak, Sunil Kumar Joshi.

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
