## [Decision Letter · Decision Letter 0]

21 Feb 2023

PONE-D-23-01914Epidemiology of unintentional childhood injuries in urban and rural areas of Nepal - A comparative studyPLOS ONE

Dear Dr. Pathak,

Thank you for submitting your manuscript to PLOS ONE. After careful consideration, we feel that it has merit but does not fully meet PLOS ONE’s publication criteria as it currently stands. Therefore, we invite you to submit a revised version of the manuscript that addresses the points raised during the review process.

ACADEMIC EDITOR: Please respnd to queries by the reviewers especially the updated references, mismatch in the number of families and the number of children included in the study. Also include definition used for including unintentional childhood injuries in the methods section. In the result section please include the numbers and denominators alongwith the percent / proportion for all the rlevant tables. These changes are must for consideration of the manuscript for acceptance. It will be better if a flow chart for sampling and data collection steps is included in the manuscript in addition to addressing the other reviewer comments. 

We look forward to receiving your revised manuscript.

Kind regards,

Hariom Kumar Solanki, M.D.

Academic Editor

PLOS ONE

Additional Editor Comments:

Dear Dr Pathak

There have been significant issues raised by the reviewers regarding methodology and/ or presentation of this manuscript. Please revise the manuscript before further consideration.

Reviewers' comments:

Reviewer's Responses to Questions

**Comments to the Author**

1. Is the manuscript technically sound, and do the data support the conclusions?

Reviewer #1: Yes

Reviewer #2: Yes

2. Has the statistical analysis been performed appropriately and rigorously? 

Reviewer #1: Yes

Reviewer #2: Yes

3. Have the authors made all data underlying the findings in their manuscript fully available?

Reviewer #1: No

Reviewer #2: Yes

4. Is the manuscript presented in an intelligible fashion and written in standard English?

Reviewer #1: No

Reviewer #2: Yes

5. Review Comments to the Author

Reviewer #1: I enjoyed reading your manuscript and this study could be the stepping stone for framing policies to prevent unintentional childhood injuries in Nepal.

having said that, I would like to point out some points that need to be addressed.

1. The abstract needs to be refined. The conclusion in the abstract is a repetition of the results.

2. Line 25 - Use recent data from WHO regarding childhood injuries.

3. Make use of references that are recent, preferably those published during the previous 10 years.

4. Line 62 - approval from the parents for any type of study is termed as "assent" and not "consent"

5. Please elaborate how the sample size of 712 was obtained

6. How the interview was conducted and was interviewed could be included in the "Methods" section.

7. Line 73 - PPS should be Probability Proportional to size

8. Please clarify how your total number of participants was 667 when your sample size was calculated to be 712.

9. For the tables, it will be proper if you provide the actual numbers besides the percentages.

10. Table 1 - Please check the socioeconomic status of the participants. The rural population seems to fare better as compared to their counterpart.

11. Have you considered fatal injuries as part of "modes of injury" for unintentional childhood injuries?

12. Line 121 - 58.4% is more than half, that is why you cannot use the term "nearly half".

13. Line 182-188 - RTIs and Burns can be addressed separately. Your study concurs with the other study you mentioned when it comes to "burn" injuries, but you mentioned that they are contrary.

14. Line 184 - You can elaborate more on this. "household chores" as you mentioned here seems to be a fire hazard.

15. Line 218 - the present study being a community based cross-sectional study utilizing convenience sampling, i would suggest that you remove the word "true" from "true snapshot".

16. Recommendations and strengths of the study can be mentioned

Reviewer #2: Attached in the reviewer comments text file. Also mentioning here

Please mention the definition of unintentional injuries used in the methods section.

Since the study population included children aged 1-16 years, who were the respondents. Was it always mothers.? What proportion of

respondents was non-mothers.?

Line 27: Can we update the citation to any recent report from WHO. I am sure there was a recent update from GBD study and also from the WHO centre for injury

prevention

Line 36: Shows the number of children that had sought for an injury. Do we have any information on treatment seeking behaviours for childhood injury

from Nepal.

Line 37 to 53 are more of content for the discussion. Can we have details on injury prevention programmes that are envisaged at WHO, or regional

level or at the country level. How evidences like those collected from your study can contribute to such a programme.? Some economic aspects of childhood

injury like school days lost, parents wages lost, etc would add a societal dimension to this public health issue. Overall background needs to be improved

a lot

Line 61: Are you sure you took consent from children aged 1-16 years.? Or was it from their parents.?

Can you provide a flowchart for the sampling technique.? Why was a convenient sampling method opted instead of a systematic random sampling. By doing so the

study appears to have been weakened a bit

The definition used for including unintentional childhood injuries is missing in the main methodology. Kindly mention and cite the reference that was

used for this definition

In the results you say there were 667 children from 658 mothers and 659 fathers. In the methods you said that only one child was chosen from a household. In

that case the number of children should match the number of mothers/fathers isn't it.?

Kindly provide the broad numbers instead of just mentioning the proportions. Please mention (n - XXX) in the table heading

Please provide Confidence interval to the prevalence estimate.

I am surprised to see 2/3rd of children suffering from falls. Guess the definition used was very broad. Did the child have any disability after all these falls.?

It's hard for me to understand the absolute number and disaggregated numbers when only the proportion is used. What does the proportion infer to.? Which

denominator was used isn't clear to me in few places especially in the multiple modes of injury table.?

Can we break Table 3 and show by each injury type (as a column). Rural vs Urban in the column head is not adding value. Suggest creating a haddon's matrix with

whatever information was collected to see what all factors were collected in your study. You may distinguish significant ones from the insignificant ones

using a * or any symbol. That would add a lot of value for a policy maker

Also let's refrain from point prevalence and shift to period prevalence as this study "COUNTS" injury events across 12 months prior to the date of interview.

6. PLOS authors have the option to publish the peer review history of their article (what does this mean?). If published, this will include your full peer review and any attached files.

Reviewer #1: No

Reviewer #2: **Yes: **Giridara Gopal Parameswaran

---

## [Author Response · Author response to Decision Letter 0]

7 Apr 2023

Response to reviewers’ comments

Reviewer #1: I enjoyed reading your manuscript and this study could be the stepping stone for framing policies to prevent unintentional childhood injuries in Nepal.

having said that, I would like to point out some points that need to be addressed.

1. The abstract needs to be refined. The conclusion in the abstract is a repetition of the results.

 Thank you for the comments. I have accepted the comment and corrected accordingly.

2. Line 25 - Use recent data from WHO regarding childhood injuries.

Thank you for the comments. I have accepted the comment and corrected accordingly.

3. Make use of references that are recent, preferably those published during the previous 10 years.

Thank you for the comments. I have accepted the comment and corrected accordingly.

4. Line 62 - approval from the parents for any type of study is termed as "assent" and not "consent"

Thank you for the comments. I have accepted the comment and corrected accordingly.

5. Please elaborate how the sample size of 712 was obtained

Sample size was calculated as follows: 

Sample size calculation was done as follows; 

Sample size (n) =(z^2.p.q)/d^2 

 Where,

 z= at 95% confidence level (standard value = 1.96) 

 p= prevalence of unintentional injury in Asian children= 5.5% 13 = 0.055

 q= 1-p = 1- 0.055 = 0.0945

 d= margin of error of 5%

 Sample Size (n) = (1.96×1.96×0.055×0.0945)/(0.05×0.05) = 80

 Considering 10% non – response rate, 

Sample size (n) = 80/0.9 = 89

 Considering the design effect, Sample size (n) = 89 X 4 = 356

 Since, the study was conducted both in rural and urban area,

 Final sample size = 356×2 = 712

Thus, the final sample size was 712 considering the prevalence of unintentional injury among Asian children as 5.5%. 13

6. How the interview was conducted and was interviewed could be included in the "Methods" section.

Thank you for the comments. I have accepted the comment and corrected accordingly.

7. Line 73 - PPS should be Probability Proportional to size

Thank you for the comments. I have accepted the comment and corrected accordingly.

8. Please clarify how your total number of participants was 667 when your sample size was calculated to be 712.

The overall response rate was 93.7% (rural: 96.0% and urban: 91.3%). Thus, the total number of participants in this study were 667 out of which 342 were from rural area and 325 from urban area.

9. For the tables, it will be proper if you provide the actual numbers besides the percentages.

Thank you for the comments. I have accepted the comment and corrected accordingly.

10. Table 1 - Please check the socioeconomic status of the participants. The rural population seems to fare better as compared to their counterpart.

Thank you for the comments. I have accepted the comment and corrected accordingly.

11. Have you considered fatal injuries as part of "modes of injury" for unintentional childhood injuries?

No, only nonfatal injuries were included.

12. Line 121 - 58.4% is more than half, that is why you cannot use the term "nearly half".

Thank you for the comments. I have accepted the comment and corrected accordingly.

13. Line 182-188 - RTIs and Burns can be addressed separately. Your study concurs with the other study you mentioned when it comes to "burn" injuries, but you mentioned that they are contrary.

Thank you for the comments. I have accepted the comment and corrected accordingly.

14. Line 184 - You can elaborate more on this. "household chores" as you mentioned here seems to be a fire hazard.

Household chores like: cooking, preparing cattle feeds, using fire for heating, etc.

15. Line 218 - the present study being a community based cross-sectional study utilizing convenience sampling, i would suggest that you remove the word "true" from "true snapshot".

Thank you for the comments. I have accepted the comment and corrected accordingly.

16. Recommendations and strengths of the study can be mentioned

Strengths 

This is a community-based study that might somewhat show the true burden of injuries as all the cases might not seek treatment in health facilities. Also, a comparison between rural and urban area has been done as the epidemiology of injuries is different in both places. 

Recommendations

Further studies are required on larger scale in order to establish the evidence base necessary for effective injury prevention program. Also, establishment of both facility-based and community-based injury registry would provide further contribution for evidence generation. Similarly, childhood injury prevention programs could be introduced at school by trained teachers and also at community level by FCHVs. Furthermore, the health systems should be strengthened to address child injuries which includes the provision of high-quality care to the injured children as well as rehabilitation and support services.

Reviewer #2: Attached in the reviewer comments text file. Also mentioning here

1. Please mention the definition of unintentional injuries used in the methods section.

The unintentional injuries are those where the intent to use the force may not necessarily intend to cause the damage. This issue can be quite complex as the intent to use force may not necessarily mean that there was an intent to cause harm or damage. Unintentional injuries are classified as per their occurrence: poisoning, burns, and scalds, drowning, falls and transport-related. 10

Thank you for the comments. I have accepted the comment and corrected accordingly.

2. Since the study population included children aged 1-16 years, who were the respondents. Was it always mothers.? What proportion of respondents was non-mothers.?

About two-third of the respondents were mothers (64.9%) followed by fathers (32.7%) and few of them were grandparents (2.4%).

3. Line 27: Can we update the citation to any recent report from WHO. I am sure there was a recent update from GBD study and also from the WHO centre for injury prevention

Thank you for the comments. I have accepted the comment and corrected accordingly.

4. Line 36: Shows the number of children that had sought for an injury. Do we have any information on treatment seeking behaviours for childhood injury from Nepal.

A study conducted in Makwanpur district of Nepal found that the use of local contacts especially FCHVs for first aid and referral purposes after an injury event were mostly preferred among the respondents. Also, delay in seeking medical care and thinking that ‘injury will heal with time’ were their common misconceptions. As the community health volunteers are not equipped with adequate medical supplies, people were bound to consult faith healers or apply home remedies such as soap water or dung water for poisoning, use of Aloe Vera for burns and application of tourniquet for snakebites.

4. Line 37 to 53 are more of content for the discussion. Can we have details on injury prevention programmes that are envisaged at WHO, or regional level or at the country level. How evidences like those collected from your study can contribute to such a programme.? Some economic aspects of childhood injury like school days lost, parents wages lost, etc would add a societal dimension to this public health issue. Overall background needs to be improved a lot.

Thank you for the comments. I have accepted the comment and corrected accordingly.

5. Line 61: Are you sure you took consent from children aged 1-16 years.? Or was it from their parents.?

Assent was taken from their parents or primary care givers

6. Can you provide a flowchart for the sampling technique.? Why was a convenient sampling method opted instead of a systematic random sampling. By doing so the study appears to have been weakened a bit

Yes, convenience sampling has weakened the study but due to the lack of sampling frame this option had to be chosen. As the study was conducted after the earthquake of 2015 in Nepal, many of the families had changed their settlement area thus it was difficult to locate them and bring into the sampling frame.

7. The definition used for including unintentional childhood injuries is missing in the main methodology. Kindly mention and cite the reference that was used for this definition

Thank you for the comments. I have accepted the comment and corrected accordingly.

8. In the results you say there were 667 children from 658 mothers and 659 fathers. In the methods you said that only one child was chosen from a household. In that case the number of children should match the number of mothers/fathers isn't it.?

The details of 658 mothers (335 and 323 in rural and urban area respectively) and 659 fathers (337 and 322 in rural and urban area respectively) were included. Total 9 mothers and 8 fathers were either separated or expired thus they were excluded from the study.

9. Kindly provide the broad numbers instead of just mentioning the proportions. Please mention (n - XXX) in the table heading

Thank you for the comments. I have accepted the comment and corrected accordingly.

10. Please provide Confidence interval to the prevalence estimate.

Out of the total 667 children, 21.7% [95% CI: 18.6 – 24.9] had sustained unintentional injuries in the past 12 months. In rural area, the prevalence was 26% [95% CI: 21.3 – 30.7] whereas in urban area it was 17.2% [95% CI: 13.1– 21.4].

11. I am surprised to see 2/3rd of children suffering from falls. Guess the definition used was very broad. Did the child have any disability after all these falls.?

Yes, this has been included in the impact of the injuries. Among the children who were injured, 22 hospitalized for treatment, 11 were left with some form of disability and 10 of them were not able to return back to school.

12. It's hard for me to understand the absolute number and disaggregated numbers when only the proportion is used. What does the proportion infer to.? Which

denominator was used isn't clear to me in few places especially in the multiple modes of injury table.?

Thank you for the comments. I have accepted the comment and corrected accordingly.

13. Can we break Table 3 and show by each injury type (as a column). Rural vs Urban in the column head is not adding value. Suggest creating a haddon's matrix with

whatever information was collected to see what all factors were collected in your study. You may distinguish significant ones from the insignificant ones

using a * or any symbol. That would add a lot of value for a policy maker

Thank you for the comments. Actually the main motive of the table is to show the scenario of the injury events as a whole rather than breaking down into each injury type as that was not the main objective of the study. Please do correct me if I am unclear regarding this.

14. Also let's refrain from point prevalence and shift to period prevalence as this study "COUNTS" injury events across 12 months prior to the date of interview.

Thank you for the comments. I have accepted the comment and corrected accordingly.

---

## [Editor Report · Decision Letter 1]

11 Apr 2023

PONE-D-23-01914R1Epidemiology of unintentional childhood injuries in urban and rural areas of Nepal - A comparative studyPLOS ONE

Dear Dr. Pathak,

Thank you for submitting your manuscript to PLOS ONE. After careful consideration, we feel that it has merit but does not fully meet PLOS ONE’s publication criteria as it currently stands. Therefore, we invite you to submit a revised version of the manuscript that addresses the points raised during the review process.

We look forward to receiving your revised manuscript.

Kind regards,

Hariom Kumar Solanki, M.D.

Academic Editor

PLOS ONE

Additional Editor Comments:

Dear Author,

It seems that the main article file attached as revised manuscript is the same as the original submission file. Please do the following

1. Make revsions to the manuscript as advised by the reviewers (or provide justification/ reason to not incorporting reviewer comments) and highlight the text where revisions have been made so that they could be easily identified by the reviewers / editors as additions / deletions from the original manuscript

2. In response to the reviewer comments, detailed changes made may not be required to be mentioned if you simply identify the place where the relevant change has been made. (For example Introduction section, line ... , so on )

Thanks and regards

---

## [Author Response · Author response to Decision Letter 1]

13 Apr 2023

Response to reviewers’ comments

Reviewer #1: I enjoyed reading your manuscript and this study could be the stepping stone for framing policies to prevent unintentional childhood injuries in Nepal.

having said that, I would like to point out some points that need to be addressed.

1. The abstract needs to be refined. The conclusion in the abstract is a repetition of the results.

 Thank you for the comments. I have accepted the comment and corrected accordingly.

2. Line 25 - Use recent data from WHO regarding childhood injuries.

Thank you for the comments. I have accepted the comment and corrected accordingly.

3. Make use of references that are recent, preferably those published during the previous 10 years.

Thank you for the comments. I have accepted the comment and corrected accordingly.

4. Line 62 - approval from the parents for any type of study is termed as "assent" and not "consent"

Thank you for the comments. I have accepted the comment and corrected accordingly.

5. Please elaborate how the sample size of 712 was obtained

Sample size was calculated as follows: 

Sample size calculation was done as follows; 

Sample size (n) =(z^2.p.q)/d^2 

 Where,

 z= at 95% confidence level (standard value = 1.96) 

 p= prevalence of unintentional injury in Asian children= 5.5% 13 = 0.055

 q= 1-p = 1- 0.055 = 0.0945

 d= margin of error of 5%

 Sample Size (n) = (1.96×1.96×0.055×0.0945)/(0.05×0.05) = 80

 Considering 10% non – response rate, 

Sample size (n) = 80/0.9 = 89

 Considering the design effect, Sample size (n) = 89 X 4 = 356

 Since, the study was conducted both in rural and urban area,

 Final sample size = 356×2 = 712

Thus, the final sample size was 712 considering the prevalence of unintentional injury among Asian children as 5.5%. 13

6. How the interview was conducted and was interviewed could be included in the "Methods" section.

Thank you for the comments. I have accepted the comment and corrected accordingly.

7. Line 73 - PPS should be Probability Proportional to size

Thank you for the comments. I have accepted the comment and corrected accordingly.

8. Please clarify how your total number of participants was 667 when your sample size was calculated to be 712.

The overall response rate was 93.7% (rural: 96.0% and urban: 91.3%). Thus, the total number of participants in this study were 667 out of which 342 were from rural area and 325 from urban area.

9. For the tables, it will be proper if you provide the actual numbers besides the percentages.

Thank you for the comments. I have accepted the comment and corrected accordingly.

10. Table 1 - Please check the socioeconomic status of the participants. The rural population seems to fare better as compared to their counterpart.

Thank you for the comments. I have accepted the comment and corrected accordingly.

11. Have you considered fatal injuries as part of "modes of injury" for unintentional childhood injuries?

No, only nonfatal injuries were included.

12. Line 121 - 58.4% is more than half, that is why you cannot use the term "nearly half".

Thank you for the comments. I have accepted the comment and corrected accordingly.

13. Line 182-188 - RTIs and Burns can be addressed separately. Your study concurs with the other study you mentioned when it comes to "burn" injuries, but you mentioned that they are contrary.

Thank you for the comments. I have accepted the comment and corrected accordingly.

14. Line 184 - You can elaborate more on this. "household chores" as you mentioned here seems to be a fire hazard.

Household chores like: cooking, preparing cattle feeds, using fire for heating, etc.

15. Line 218 - the present study being a community based cross-sectional study utilizing convenience sampling, i would suggest that you remove the word "true" from "true snapshot".

Thank you for the comments. I have accepted the comment and corrected accordingly.

16. Recommendations and strengths of the study can be mentioned

 Thank you for the comments. Strengths and recommendations have been added in the manuscript.

Reviewer #2: Attached in the reviewer comments text file. Also mentioning here

1. Please mention the definition of unintentional injuries used in the methods section.

The unintentional injuries are those where the intent to use the force may not necessarily intend to cause the damage. This issue can be quite complex as the intent to use force may not necessarily mean that there was an intent to cause harm or damage. Unintentional injuries are classified as per their occurrence: poisoning, burns, and scalds, drowning, falls and transport-related. 10

Thank you for the comments. I have accepted the comment and corrected accordingly.

2. Since the study population included children aged 1-16 years, who were the respondents. Was it always mothers.? What proportion of respondents was non-mothers.?

About two-third of the respondents were mothers (64.9%) followed by fathers (32.7%) and few of them were grandparents (2.4%).

3. Line 27: Can we update the citation to any recent report from WHO. I am sure there was a recent update from GBD study and also from the WHO centre for injury prevention

Thank you for the comments. I have accepted the comment and corrected accordingly.

4. Line 36: Shows the number of children that had sought for an injury. Do we have any information on treatment seeking behaviours for childhood injury from Nepal.

A study conducted in Makwanpur district of Nepal found that the use of local contacts especially FCHVs for first aid and referral purposes after an injury event were mostly preferred among the respondents. Also, delay in seeking medical care and thinking that ‘injury will heal with time’ were their common misconceptions. As the community health volunteers are not equipped with adequate medical supplies, people were bound to consult faith healers or apply home remedies such as soap water or dung water for poisoning, use of Aloe Vera for burns and application of tourniquet for snakebites.

4. Line 37 to 53 are more of content for the discussion. Can we have details on injury prevention programmes that are envisaged at WHO, or regional level or at the country level. How evidences like those collected from your study can contribute to such a programme.? Some economic aspects of childhood injury like school days lost, parents wages lost, etc would add a societal dimension to this public health issue. Overall background needs to be improved a lot.

Thank you for the comments. I have accepted the comment and corrected accordingly.

5. Line 61: Are you sure you took consent from children aged 1-16 years.? Or was it from their parents.?

Assent was taken from their parents or primary care givers

6. Can you provide a flowchart for the sampling technique.? Why was a convenient sampling method opted instead of a systematic random sampling. By doing so the study appears to have been weakened a bit

Yes, convenience sampling has weakened the study but due to the lack of sampling frame this option had to be chosen. As the study was conducted after the earthquake of 2015 in Nepal, many of the families had changed their settlement area thus it was difficult to locate them and bring into the sampling frame.

7. The definition used for including unintentional childhood injuries is missing in the main methodology. Kindly mention and cite the reference that was used for this definition

Thank you for the comments. I have accepted the comment and corrected accordingly.

8. In the results you say there were 667 children from 658 mothers and 659 fathers. In the methods you said that only one child was chosen from a household. In that case the number of children should match the number of mothers/fathers isn't it.?

The details of 658 mothers (335 and 323 in rural and urban area respectively) and 659 fathers (337 and 322 in rural and urban area respectively) were included. Total 9 mothers and 8 fathers were either separated or expired thus they were excluded from the study.

9. Kindly provide the broad numbers instead of just mentioning the proportions. Please mention (n - XXX) in the table heading

Thank you for the comments. I have accepted the comment and corrected accordingly.

10. Please provide Confidence interval to the prevalence estimate.

Out of the total 667 children, 21.7% [95% CI: 18.6 – 24.9] had sustained unintentional injuries in the past 12 months. In rural area, the prevalence was 26% [95% CI: 21.3 – 30.7] whereas in urban area it was 17.2% [95% CI: 13.1– 21.4].

11. I am surprised to see 2/3rd of children suffering from falls. Guess the definition used was very broad. Did the child have any disability after all these falls.?

Yes, this has been included in the impact of the injuries. Among the children who were injured, 22 hospitalized for treatment, 11 were left with some form of disability and 10 of them were not able to return back to school.

12. It's hard for me to understand the absolute number and disaggregated numbers when only the proportion is used. What does the proportion infer to.? Which

denominator was used isn't clear to me in few places especially in the multiple modes of injury table.?

Thank you for the comments. I have accepted the comment and corrected accordingly.

13. Can we break Table 3 and show by each injury type (as a column). Rural vs Urban in the column head is not adding value. Suggest creating a haddon's matrix with

whatever information was collected to see what all factors were collected in your study. You may distinguish significant ones from the insignificant ones

using a * or any symbol. That would add a lot of value for a policy maker

Thank you for the comments. Actually, the main motive of the table is to show the scenario of the injury events as a whole rather than breaking down into each injury type as that was not the main objective of the study. Please do correct me if I am unclear regarding this.

14. Also let's refrain from point prevalence and shift to period prevalence as this study "COUNTS" injury events across 12 months prior to the date of interview.

Thank you for the comments. I have accepted the comment and corrected accordingly.

---

## [Decision Letter · Decision Letter 2]

6 Jun 2023

Epidemiology of unintentional childhood injuries in urban and rural areas of Nepal - A comparative study

PONE-D-23-01914R2

Dear Dr. Pathak,

We’re pleased to inform you that your manuscript has been judged scientifically suitable for publication and will be formally accepted for publication once it meets all outstanding technical requirements.

Kind regards,

Hariom Kumar Solanki, M.D.

Academic Editor

PLOS ONE

Additional Editor Comments (optional):

Reviewers' comments:

Reviewer's Responses to Questions

**Comments to the Author**

1. If the authors have adequately addressed your comments raised in a previous round of review and you feel that this manuscript is now acceptable for publication, you may indicate that here to bypass the “Comments to the Author” section, enter your conflict of interest statement in the “Confidential to Editor” section, and submit your "Accept" recommendation.

Reviewer #3: All comments have been addressed

2. Is the manuscript technically sound, and do the data support the conclusions?

Reviewer #3: Yes

3. Has the statistical analysis been performed appropriately and rigorously? 

Reviewer #3: Yes

4. Have the authors made all data underlying the findings in their manuscript fully available?

Reviewer #3: Yes

5. Is the manuscript presented in an intelligible fashion and written in standard English?

Reviewer #3: Yes

6. Review Comments to the Author

Reviewer #3: The authors have addressed all comments in the draft. The article right now appears to be in much better shape than before. Accepted

7. PLOS authors have the option to publish the peer review history of their article (what does this mean?). If published, this will include your full peer review and any attached files.

Reviewer #3: No

---

## [Editor Report · Acceptance letter]

13 Jun 2023

PONE-D-23-01914R2 

Epidemiology of unintentional childhood injuries in urban and rural areas of Nepal- A comparative study 

Dear Dr. Pathak:

I'm pleased to inform you that your manuscript has been deemed suitable for publication in PLOS ONE. Congratulations! Your manuscript is now with our production department. 

Kind regards, 

on behalf of

Dr. Hariom Kumar Solanki 

Academic Editor

PLOS ONE